# Glycolytic Plasticity of Metastatic Lung Cancer Captured by Noninvasive ^18^F-FDG PET/CT and Serum ^1^H-NMR Analysis: An Orthotopic Murine Model Study

**DOI:** 10.3390/metabo13010110

**Published:** 2023-01-09

**Authors:** Yi-Hsiu Chung, Tsai-Hsien Hung, Ching-Fang Yu, Cheng-Kun Tsai, Chi-Chang Weng, Fujie Jhang, Fang-Hsin Chen, Gigin Lin

**Affiliations:** 1Department of Medical Research and Development, Chang Gung Memorial Hospital at Linkou, Taoyuan 333423, Taiwan; 2Institute of Stem Cell and Translational Cancer Research, Chang Gung Memorial Hospital at Linkou, Taoyuan 333423, Taiwan; 3Radiation Biology Research Center, Institute for Radiological Research, Chang Gung Memorial Hospital at Linkou, Chang Gung University, Taoyuan 333323, Taiwan; 4Clinical Metabolomics Core Lab, Chang Gung Memorial Hospital at Linkou, Taoyuan 333423, Taiwan; 5Department of Medical Imaging and Radiological Sciences, Chang Gung University, Taoyuan 333323, Taiwan; 6Institute of Nuclear Engineering and Science, National Tsing Hua University, Hsinchu 300044, Taiwan; 7Department of Medical Imaging and Intervention, Institute for Radiological Research, Chang Gung Memorial Hospital at Linkou, Chang Gung University, Taoyuan 333323, Taiwan

**Keywords:** lung cancer, ^18^F-FDG, 1H-NMR, animal study, metastasis

## Abstract

We aim to establish a noninvasive diagnostic platform to capture early phenotypic transformation for metastasis using ^18^F-FDG PET and ^1^H-NMR-based serum metabolomics. Mice with implantation of NCI-H460 cells grew only primary lung tumors in the localized group and had both primary and metastatic lung tumors in the metastatic group. The serum metabolites were analyzed using ^1^H-NMR at the time of PET/CT scan. The glycolysis status and cell proliferation were validated by Western blotting and staining. A receiver operating characteristic (ROC) curve analysis was performed to evaluate the diagnostic accuracy of SUV_mean_ and serum metabolites in metastasis. In the metastatic mice, the SUV_mean_ of metastatic tumors was significantly higher than that of primary lung tumors in PET images, which was supported by elevated glycolytic protein expression of HK2 and PKM2. The serum pyruvate level in the metastatic group was significantly lower than that in the localized group, corresponding to increased pyruvate-catalyzed enzyme and proliferation rates in metastatic tumors. In diagnosing localized or metastatic tumors, the areas under the ROC curves of SUV_mean_ and pyruvate were 0.92 and 0.91, respectively, with *p* < 0.05. In conclusion, the combination of ^18^F-FDG PET and ^1^H-NMR-based serum metabolomics demonstrated the feasibility of a glycolytic platform for diagnosing metastatic lung cancers.

## 1. Introduction

Lung cancer is the leading cause of cancer death among both men and women, accounting for almost 25% of all cancer deaths in American Cancer Society reports [1]. The 5-year survival rate is 63% in patients with localized lung cancer, but it decreases to 35% if the cancer spreads to nearby tissue, and drops to approximately 7% if distant metastasis occurs [2]. Primary tumors and metastases show a variety of metabolic processes that are tightly linked to the characteristics of the tumor microenvironment [3]. More importantly, based on the various treatment responses of primary, metastatic, or recurrent tumors, understanding the metabolic status of the lesions would be a good means for predicting treatment responses [4].

Positron emission tomography (PET) is a noninvasive imaging tool for cancer diagnosis, staging, and predicting prognosis [5,6]. ^18^F-fluorodeoxyglucose (FDG) is a glucose analog PET tracer that can be used to evaluate the metabolic status of tumors [7]. It has been reported that FDG PET/CT provides specific molecular and anatomic three-dimensional images for primary lung tumors and metastasis, such as contralateral lung, liver, bone, and lymph node metastases, in preclinical studies and clinical use [5,8]. FDG only partially provides macroinformation on glycolysis in primary tumors and metastasis, and poor CT image contrast is not suitable for metastasis in preclinical studies. Once FDG is transported by the glucose transporters, after phosphorylation by hexokinase, however, FDG-6-phosphate cannot proceed along the glycolytic pathway. Through the FDG images, we only know the upstream glycolytic status of the tumors [9], but for further glycolysis of tumors, for example, lactate and pyruvate, the end products of glycolysis cannot be investigated by FDG.

A new approach was prosperously exploited in metabolomics and nuclear magnetic resonance (NMR), in the past decade. The ^1^H-NMR metabolomic approach has emerged as a considerable tool to globally profile tumor metabolites [10] to gain insights into how tumors orchestrate metabolic alterations in tumor progression and potential therapeutic means. Pan et al. reported meaningful changes in metabolic profiles from serum metabolomic analysis in human lung cancer-bearing mice compared with those treated with anlotinib [11]. Potential biomarkers such as lactate [12], glutamate [12], and glutamine [12,13] from serum ^1^H-NMR metabolomics studies for lung cancer patients exist in the literature, but a preclinical serum metabolomic study in orthotopic human lung cancer mice associated with various metastases appeared to have not been explored.

Our hypothesis of glycolytic metabolite alteration in the progression of metastasis is that glycolysis is elevated and the tricarboxylic acid (TCA) cycle is activated to produce adenosine triphosphate (ATP) and metabolites that metastatic tumor cells need for expansion and invasion. At the same time, the serum glycolytic-related metabolites released into the blood circulatory system are decreased due to the catalysis of products of glycolysis in tumor cells. Figure 1 illustrates the hypothesis of glycolytic metabolite alterations in metastatic progression.

In this study, we used ^18^F-FDG PET/CT to quantify the analog glucose uptake in primary lung tumors and metastatic tumors in a lung cancer mouse model. Concurrently, we performed serum metabolism analysis of tumor-bearing mice to identify feasible biomarkers for the murine metastasis model. Therefore, the aim of this study was to investigate the combination of quantitative metabolic PET images and serum metabolite analysis to lend further support to the hypothesis that tumor glycolysis related to systemic circulation plays a critical role in the diagnosis and staging of lung cancer.

## 2. Materials and Methods

### 2.1. Cell Lines

The human NCI-H460 (BCRC 60373) large cell carcinoma cell line was purchased from Bioresource Collection and Research Center (Hsinchu, Taiwan). NCI-H460 cells were maintained in RPMI-1640 medium containing 10% fetal bovine serum and 1% penicillin/streptomycin (all reagents from GIBCO, Thermo Fisher Scientific, Inc., Waltham, MA, USA) and incubated at 37 °C in humidified air containing 5% CO_2_.

### 2.2. In Vivo and Ex Vivo Experimental Design

The mice were subjected to surgical orthotopic lung tumor implantation 3 weeks before imaging. The body weight of the mice was recorded twice per week until sacrifice. PET/CT imaging was performed at week 3 and week 5 after implantation. The sera of mice were collected by cardiac puncture before sacrifice. The tumor and surrounding tissues were removed and harvested for further pathology validation. The experimental protocols were in accordance with the animal guidelines set out and approved by the Institutional Animal Care and Use Committee of Chang Gung Memorial Hospital, Taiwan (IACUC 2020041002). The experimental flow chart is shown in Figure 2.

### 2.3. Surgical Procedure for Orthotopic Lung Cancer

Cultured NCI-H460 (1 × 10^4^) cells were harvested and suspended in RPMI-1640 medium. Before injection, the cells were mixed with Matrigel^®^ Basement Membrane Matrix Growth Factor Reduced (GFR) (Corning, NY, USA) at a 1:1 ratio and kept on ice until injection. For injection, 2 μL of the mixed cell suspension was loaded into a 10 μL syringe (Hamilton, NV, USA) with a 26S-gauge needle. The syringe was positioned on the modified stereotactic injection system. Mice were anesthetized with 2% isoflurane. The mice were positioned in the lateral decubitus position with the left chest facing up thereafter. An approximately 1 cm incision was made over the skin below the scapula with surgical scissors. Gently, the chest wall muscles, myolemma, or other fascial tissues were spread until the intercostal space and rib were clearly visible and the pleura and lung were pink in color. Mice were moved to the stereotactic injection system stage (KDS-310-PLUS, Holliston, MA, USA). The cell-loaded syringe was gently adjusted until its tip finely touched the intercostal space. With a stereotactic injection system setup, the syringe was advanced 3.5 mm into the lung parenchyma and 2 μL of tumor cell mixture was injected at a 0.5 μL/minute flow rate. After injection of cells, the syringe was kept still for 5 min before being withdrawn from the tissues. Then, the covering muscles and skin of mice were sutured.

### 2.4. In Vivo ^18^F-FDG PET/CT Imaging

Before performing FDG PET, to reduce the myocardial FDG uptake, the food pellets of mice were changed from a standard diet to sunflower seeds in 48 h of ad libitum water or fasted overnight [14]. For FDG PET imaging, tumor-bearing mice were injected with 8.0 ± 0.33 MBq of ^18^F-FDG via tail vein and contained in the chamber for a 60 min uptake period under 1–2% isoflurane anesthesia conditions with an infrared heat lamp to prevent hypothermia. Then, the mice were subjected to a 10-min image acquisition in the supine position using a microPET-CT scanner (nanoScan PET/CT, Mediso, Hungary). During imaging, mice were anesthetized with 2% isoflurane and placed near the center of the field of view. The scan bed kept the temperature at 32 °C by air flow warming. PET images were subjected to attenuation correction by CT images. All images were reconstructed by the 3D ordered-subsets expectation maximization iterative method. The voxel size of PET images was 0.4 × 0.4 × 0.4 mm with field of view. The unit of PET images was expressed as standardized uptake values (SUVs), in which the radioactivity concentrations in the image were multiplied by the individual body weight and divided by the injected dose. The tumor regions were determined semiautomatically using a threshold of 20% of the maximum SUV minus the minimum SUV from manually contoured regions of interest (ROIs). FDG uptake in tumors was expressed as SUV_mean_. All images were analyzed using PMOD version 4.1 (PMOD Techologies LLC, Fällanden, Switzerland)

### 2.5. Serum Extraction and ^1^H-NMR Metabolism Analysis

Serum extraction was made by standard protocol [10]. Serum samples were diluted with an equivalent volume of prepared buffer (0.075 M Na_2_HPO_4_, 0.08% TSP, 2 mM NaN_3_, 20% D_2_O in ddH_2_O, pH 7.4) for vortexing for 5 s. The mixing samples were centrifuged at 12,000 rpm at 4 °C for 30 min. An aliquot of 150 μL of mixing sample was slowly transferred into a 4-inch SampleJet NMR tube without any air bubbles for further NMR study [15].

^1^H-NMR metabolism analysis was conducted using a Bruker Avance III HD 600 MHz spectrometer (Bruker Biospin GmbH, Rheinstetten, Germany) operating with a 14.1 T magnet (600 MHz) and equipped with a 5 mm inverse triple resonance CryoProbe with cold preamplifiers for ^1^H with a z-axis gradient with automated tuning and matching [16,17]. The spectrometer containing the samples was kept at 297 K [18].

Acquisition was carried out using Bruker cpmgpr1d, a widely used pulse sequence for serum NMR analysis [10,17]. Briefly, the continuous wave irradiation was at the water frequency using 25 Hz RF strength during the relaxation delay (4 s). The magnetic field z-gradient was applied for 1 ms, and the acquisition time was 2.7 s. The spectral window was set to 20 ppm for serum samples. The 32 transients were acquired with 96 k data points for serum. The receiver gain was set to 90.5. The spin-echo delay is set at 300 μs and a minimum of 128 loops [17]. Our platform used a standard internal reference sodium 3-trimethylsilyl-2,2,3,3-tetradeuteropropionate (TSP) to calibrate metabolic concentrations in the aqueous phase [19,20]. The aqueous metabolites were identified by Chenomx NMR Suite software (Chenomx, Inc., Edmonton, AB, Canada). All metabolite data were analyzed by MetaboAnalyst 5.0 [21]. A statistical analysis (one factor) module was performed with normalization by median, log transformation, and Pareto scaling in data processing for this dataset that is approximate to a normal distribution shape [22]. Metabolites satisfying three conditions: fold change over 1.2 or less than 0.8 [23], *p*-value in Wilcoxon rank-sum test less than 0.05 for a small number of samples, and not a full normal distribution of samples and variable importance in projection (VIP) score >1.5 in the PLS-DA model were considered significant [19].

### 2.6. Ex Vivo Histopathological Assays of the Proliferation Rates and Glycolysis in Primary Tumors and Metastases

Tumor tissues were embedded in OCT medium (Sakura Finetek, Torrance, CA, USA) and stored at −80 °C after harvesting. Frozen sections (thickness 20 μm) were fixed with cold 100% methanol for 5 min, washed twice with phosphate-buffered saline (PBS), and blocked with PBS containing 1% bovine serum albumin and 0.01% Tween-20 for 1 h at room temperature to reduce nonspecific binding. Sections were then incubated overnight at 4 °C with the purified antibodies against specific markers: anti-Ki67 (Abcam, Cambridge, UK). Fluorescent dye-conjugated secondary antibodies (Invitrogen, Carlsbad, CA, USA) bound to primary antibodies were added for 1 h at room temperature. The adjacent slices of tumors and tissues were also subjected to hematoxylin and eosin staining [24]. Images were captured and analyzed on a HistoFAXS Tissue Analysis System (TissueGnostics, Wien, Austria).

Tumors were lysed before Western blotting analysis. The tumor samples were washed with PBS, cut into small pieces, and then lysed with PierceTM RIPA Buffer (Thermo Fisher Scientific, Inc., Waltham, MA, USA) supplemented with protease inhibitor cocktail (Roche, Mannheim, Germany) on ice for 15 min. Tumor lysates were centrifuged at 14,000× *g* for 15 min at 4 °C, and the supernatants were harvested as protein samples. The total protein content was determined using the Coomassie Plus^TM^ Protein Assay Reagent (Thermo Fisher Scientific, Inc., Waltham, MA, USA). Tumor lysates were analyzed by Western blotting following the manufacturer’s protocol. Tumor lysate protein was transferred onto PVDF membranes (PerkinElmer, Waltham, MA, USA). After transfer, the membranes were blocked in 5% nonfat milk in double-distilled water for 1 h at room temperature (RT). In a previous report, glucose transporter type 1 and hexokinase activity as well as pyruvate kinase M2 (PKM2) were associated with metabolic reprogramming in lung cancer NCI-H460 cells [25]. Furthermore, pyruvate carboxylase (PC) catalyzes the irreversible carboxylation of pyruvate to form oxaloacetate (OAA) in the TCA cycle [26]. Then, the blots were incubated with anti-glucose transporter 1 (Glut1, Abcam, Cambridge, UK), anti-hexokinase 1 (HK1, Proteintech, Chicago, IL, USA), anti-hexokinase 2 (HK2, Abcam, Cambridge, UK), anti-pyruvate kinase M2 (PKM2, Cell Signaling Technology, Danvers, MA, USA) and anti-pyruvate carboxylase (PC, Santa Cruz, CA, USA) overnight at 4 °C. GAPDH was used as a loading control. The membranes were then incubated with horseradish peroxidase-conjugated goat anti-rabbit antibody (AbRay, New Taipei City, Taiwan) for targeted antibodies and goat anti-mouse IgG (H+L) antibodies (AbRay, New Taipei City, Taiwan) for the loading control for 1 h at RT. The signal was measured through chemiluminescence detection with Ultra ECL-HRP Substrate solution (Thermo Fisher Scientific, Inc., Waltham, MA, USA), visualized and photographed using an Amersham Imager 600 Imaging System (GE HealthCare, Chicago, IL, USA), and analyzed using ImageJ image processing software. Three tumors from each group were included.

### 2.7. Statistical Analysis

All data are expressed as the means ± standard deviations (SD) except for the NMR results. In vivo FDG uptake and histological results of the primary lung tumors and metastases were compared with the nonparametric Mann-Whitney U test. Data were obtained from ^1^H-NMR metabolomics experiments using the nonparametric Wilcoxon rank-sum test. The data are expressed as the means ± standard error of the mean (SEM). A ROC curve analysis was performed on evaluation of the PET imaging and serum metabolite results. ROC curves were generated, and the areas under the curve (AUCs) were calculated to compare the value of SUV_mean_ and normalized pyruvate concentration for metastasis prediction (localized mice, N = 7 and metastatic mice, N = 12). All analyses were carried out in GraphPad Prism 6 (GraphPad Inc., San Diego, CA, USA). Statistical significance was determined by a *p* < 0.05.

## 3. Results

### 3.1. ^18^F-FDG PET/CT Detects Primary Tumors and Metastases

Glucose metabolism in primary tumors and metastases was examined using ^18^F-FDG PET in vivo imaging. ^18^F-FDG PET and CT images were obtained through whole-body scanning, and fused tomographic images were generated (Figure 3a,b). Here, we defined metastatic tumors as tumors located in the contralateral thorax and right lung parenchyma. The ^18^F-FDG uptake of primary lung tumors between the localized and metastatic groups showed no significant difference (localized group, SUV_mean_: 1.10 ± 0.34, N = 6, vs. metastatic group, SUV_mean_: 1.20 ± 0.35, N = 5, *p* > 0.79, Figure 3c). However, in the metastatic group, ^18^F-FDG uptake in metastatic tumors was significantly elevated compared with that in primary tumors (primary tumors, SUV_mean_: 1.20 ± 0.35 vs. metastatic tumors, SUV_mean_: 1.97 ± 0.30, N = 5, *p* < 0.01, Figure 3d). Comparing various metastatic tumors with primary tumors, the metastatic right pleural tumors and the lower pleural tumors showed increased ^18^F-FDG uptake, as shown in Figure 3e (primary tumors, SUV_mean_: 1.21 ± 0.39, N = 16, right pleural tumors, SUV_mean_: 2.21 ± 0.60, N = 13, *p* < 0.001, lower pleural tumors, SUV_mean_: 1.65 ± 0.46, N = 10, *p* > 0.05). However, the left pleural tumors, which were diffused or transmitted from the left lung parenchyma, showed significantly increased uptake of ^18^F-FDG compared with primary lung tumors (SUV_mean_: 2.17 ± 0.72, N = 12, *p* < 0.001).

### 3.2. Metabolites in Serum by ^1^H-NMR Analysis

The ^1^H-NMR analysis identified aqueous metabolites including acetate, alanine, choline, citrate, creatine, formate, fumarate, glucose, glutamine, isoleucine, lactate, leucine, mannose, methionine, o-phosphocholine, phenylalanine, pyruvate, threonine, tyrosine, urea, and valine. Table 1 shows the analyzed metabolites with VIP scores over 1.5, including pyruvate, o-phosphocholine, phenylalanine, creatine, and fumarate. Notably, the level of the metabolite pyruvate in serum was significantly lower in the metastatic group than in the localized group (pyruvate, VIP score = 2.34, fold-changes (FC) of metastasis to the localized group: 0.66, Wilcoxon rank-sum tests, *p* = 0.004). The representative serum NMR spectra from the localized group and metastatic group show the concentrations of various metabolites (Figure 4a). Quantification of the processed pyruvate serum levels in the localized group and metastatic group is shown in Figure 4b. There was a slightly lower original pyruvate level in the serum of the metastatic group (localized group, pyruvate level: 626.2 ± 121, N = 8 vs. metastatic group, pyruvate level: 404.2 ± 51.12, N = 12, *p* = 0.07). With the procession and normalization by MetaboAnalyst software, the significantly lower normalized pyruvate level in the serum of the metastatic group is presented in Figure 3c (localized group, pyruvate level: 0.29 ± 0.1, N = 8 vs. metastatic group, pyruvate level: −0.19 ± 0.1, N = 12, *p* < 0.01). Furthermore, o-phosphocholine levels in serum were increased in the metastatic group (VIP score = 1.84, FC = 1.46, *p* = 0.057). Phosphocholine is an intermediate in the synthesis of phosphatidylcholine in cell membranes and is converted from ATP and choline [27], indicating the activation of energy conversion in the metastatic group. Moreover, phenylalanine levels in serum were decreased in the metastatic group (VIP score = 1.74, FC = 0.75, *p* = 0.057). Phenylalanine is an essential amino acid for the synthesis of proteins the body needs, implying that a large amount of phenylalanine might be utilized for the synthesis of proteins in cell proliferation in the metastatic group [28]. Fumarate, the metabolite of the TCA cycle in mitochondria, showed a lower concentration in the serum of the metastatic group (VIP score = 1.54, FC = 0.80, *p* = 0.343). In addition, compared with normal mice, certain metabolites in serum showed significant changes in the tumor-bearing mice, listed in Appendix A.

### 3.3. Elevated Proliferation Rates and Glycolysis in Metastatic Tumors

To understand the detailed changes in the cellular system between primary tumors and metastatic tumors, we conducted H&E staining, immunofluorescence staining of Ki-67, and Western blotting of Glut1, HK1, HK2, PKM2, and PC. Figure 5 shows the H&E staining and Ki-67 fluorescent staining of primary tumors and metastatic tumors. The elevated proliferated signals of Ki-67 fluorescence were revealed in metastatic tumors corresponding to poor differentiation of tumor cells in H&E staining compared with primary tumors. The quantitative expression of fluorescent Ki-67 in metastatic tumors was significantly increased compared to that in primary lung tumors (primary lung tumors 6.052 ± 1.934% vs. metastatic tumors 13.41 ± 1.184%, N = 3 for each group, *p* < 0.05). In the Western blot study, the quantification of Glut1, HK1, HK2, PKM2, and PC protein expression was normalized to GAPDH expression. The significantly increased HK2, PKM2, and PC protein expression in the metastatic tumors is presented in Figure 6 and Appendix A (the ratio of HK2 to GAPDH, primary tumor = 0.40 ± 0.08 vs. metastatic tumor = 0.73 ± 0.23, N = 3 for each group, *p* < 0.05; the ratio of PKM2 to GAPDH, primary tumor = 0.82 ± 0.29 vs. metastatic tumor = 1.50 ± 0.18, n = 4 for each group, *p* < 0.05; the ratio of PC to GAPDH, primary tumor = 0.42 ± 0.26 vs. metastatic tumor = 0.73 ± 0.34, N = 5 for each group, *p* < 0.05). However, no significant differences in Glut1 and HK1 protein expression between primary tumors and metastatic tumors were found.

### 3.4. Outstanding AUC Values of 18F-FDG PET/CT Imaging Parameters and Serum Pyruvate Levels in ROC Analysis

The importance of the valued biomarkers for the prediction of metastases has received attention. We took advantage of SUV_mean_ in tumor uptake from ^18^F-FDG PET and normalized pyruvate serum levels as biomarkers for the diagnosis of metastases. ROC curves were plotted based on the SUV_mean_ of tumors and normalized pyruvate serum levels of localized and metastatic tumor mice. Figure 7 shows the ROC curves for evaluation of the diagnostic accuracy of SUV_mean_ and normalized serum pyruvate levels in the differentiation of metastases from localized tumors. The AUCs of SUV_mean_ and normalized pyruvate levels were 91.7% (SUV_mean_, 95% CI, 0.78–1.06, best cutoff value 1.68, sensitivity = 83.3% and specificity = 83.3%) and 91.1% (normalized pyruvate levels, 95% CI, 0.77–1.05, best cutoff value 0.224, sensitivity = 75% and specificity = 62.5%) with logistic regression, *p* < 0.01, respectively. Two of the false-negative metastasis tumors made by SUV of PET were correctly diagnosed according to metabolomics analysis by pyruvate. One of the false-negative primary groups made by pyruvate was correctly diagnosed based on PET. These exemplified the complementary roles of PET and metabolomics approaches in this platform.

## 4. Discussion

The present study was an initial preclinical study using a combination of imaging-based tumor glycolysis and serum metabolite levels concurrently in differential diagnoses of lung cancer metastases. The advantages of the proposed platform are to screen many metabolites from serum and find a significant biomarker, pyruvate, which is related to glycolysis. Furthermore, we used PET images to map the glycolysis of the tumor. Through this platform, we know that glycolysis is not only in the tumor itself but also affects the whole body. Higher ^18^F-FDG uptake was demonstrated in metastatic tumors compared to primary tumors, supported by the significantly increased metastatic tumor protein expression of HK2, which is relevant to the upstream step of glycolysis. In addition, we also found increased metastatic tumor protein expression of PKM2, which catalyzes the conversion of phosphoenolpyruvate to pyruvate, the final step of glycolysis. Additionally, significantly increased metastatic tumor protein expression of PC were indicating the catalyzation of pyruvate in the TCA cycle was strongly promoted. The active TCA cycle in metastatic tumors led to significantly increased levels of Ki-67 proliferation fluorescent staining signals. Significantly lower serum pyruvate levels were found in metastatic mice, which may be further related to catalysis by pyruvate carboxylase in the TCA cycle.

Some imaging techniques have been used for monitoring lung cancer progression such as ultrasound imaging [29] or PET/CT [30]. Luciferase-transfected cancer cells with bioluminescent techniques are widely used to identify the success of lung cancer mouse models [31], but the disadvantages of bioluminescence are an inefficient transfection rate as well as one-dimensional images. Furthermore, fused PET/CT images, metastases in the lung and thorax, even in the abdomen and liver, could be detected more precisely and accurately by PET/CT [32]. Although chest computed tomography (CT) is the most effective imaging tool for diagnosing primary lung cancer in clinical practice and preclinical studies [33], technical constraints still exist in detecting small metastases. These exemplified the complementary roles of PET and metabolomics approaches in this platform.

In our study, we found that the higher expression of HK2 in metastatic tumors, where tumor glycolysis increased, supported the elevated FDG uptake of tumors. In line with our study, the pyruvate kinase M2 (PKM2) catalyzing the formation of pyruvate from phosphoenolpyruvate in the last step of glycolysis [34], was a potential serum biomarker for promoting tumor progression in lung cancer cells and patients [35]. Furthermore, pyruvate undergoes two further metabolic pathways in cells. One is conversion to lactate under anaerobic conditions, and the other is entering the TCA cycle under aerobic conditions. Christen et al. [36] and Kiesel et al. [26] both reported that pyruvate carboxylase (PC), which catalyzes pyruvate to oxaloacetate (OAA) in the TCA cycle, plays a key role in the lung metastatic process in breast cancer, which was consistent with this lung cancer model. The significantly increased expression of PC indicates the TCA cycle is very active, corresponding to the elevated proliferation rate [37]. Overall, our hypothesis is that the produced pyruvate from PKM2 is highly utilized in the TCA cycle in metastatic tumor cells; herein, it leads to a reduced pyruvate level in the circulatory system.

With serum metabolites information applied in the future, the accuracy of diagnosing lung cancer is expected to increase. In our study, SUV_mean_ from ^18^F-FDG PET and normalized pyruvate serum levels assessed in this study are expected to be potential metabolic biomarkers for the diagnosis of lung metastases in clinical use. One study agreed with our proposed strategy. Lin et al. showed that the prognosis of patients with limited disease in small cell lung cancer (SCLC) who had a low SUV_max_ of the primary tumor from PET studies and normal serum lactate dehydrogenase (LDH) was better than those with a high SUV_max_ and/or high LDH [38]. The use of FDG PET images to discover the heterogeneous metabolic response to treatment in patients with NSCLC found patients with a high metabolic response are significantly associated with poor prognosis [39]. The additional kinetic parameters such as K1, k2, Ki, and Ki/K1 in the dynamic PET scan, compared with SUV_max_, were associated with not only the metabolic condition of the lesion but also its blood flow and microenvironment [40], in order to differentiate metastatic and non-metastatic LNs. It appears that the combination of imaging biomarkers and serum metabolites or protein levels could become a feasible tool in the prediction of the prognosis of patients with lung cancers.

The limitation of the present study needs to be addressed here. This platform has only been conducted using large-cell lung cancer. Other cell types such as adenocarcinoma, squamous cell carcinoma, and small cell lung cancer, may cause a different diagnosis. For more completed lung cancer studies, we need to use this platform in various lung cancer types in further studies. Moreover, pneumonia or other pulmonary inflammation in FDG images shows increased uptake in lesions [41], which might cause a false-positive rate to diagnose lung cancer, which warrants further study. Furthermore, machine learning indeed has the potential to improve the accuracy and validation of statistical measurement. However, the limited number of PET image data in the present study seems a constraint for building a model. Machine learning could be applied for metabolites analysis in future studies with larger data sets.

## 5. Conclusions

The metastatic tumors showed elevated ^18^F-FDG uptake correlated with increased cellular glycolysis. The increased pyruvate carboxylase and high proliferation rates of metastatic tumors could be associated with the promoted utilization of pyruvate, which led to the decreased serum pyruvate level found in metastatic mice. The combination of ^18^F-FDG PET/CT and ^1^H-NMR-based serum metabolomics demonstrated the feasibility of a metabolic platform for the differentiation of metastatic lung cancers from localized ones. In the future, this platform could potentially provide metabolic information for therapeutic strategies in lung cancer.

## Figures and Tables

**Figure 1 metabolites-13-00110-f001:**
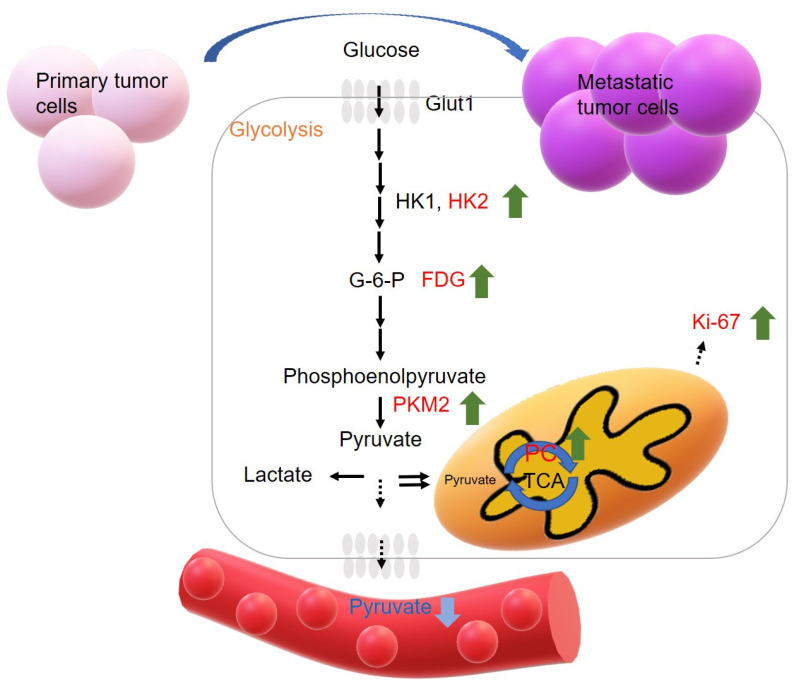
The hypothesis of metabolism alternation in the progression of metastasis. Tumor needs to acquire a large amount of energy for expansion and invasion. Elevated glycolysis and active TCA (tricarboxylic acid) cycle produce ATP (adenosine triphosphate) and metabolites that metastatic tumor cells need. Meanwhile, the metabolites, pyruvate released from the cell into the circulatory system would be decreased due to the catalyzation of pyruvate in tumor cells.

**Figure 2 metabolites-13-00110-f002:**
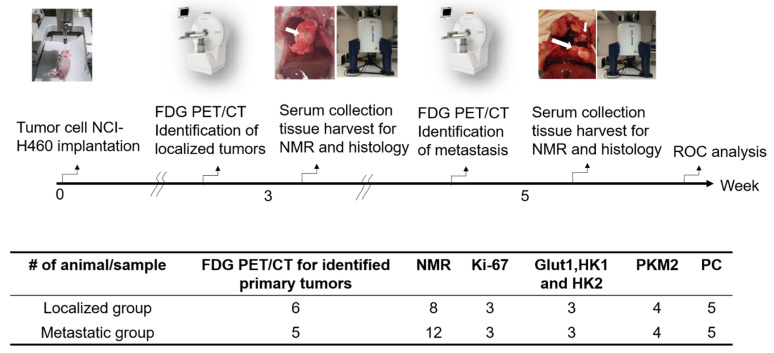
Flow chart of animal experiments. The number of animals used in each experiment are displayed. Mice were identified localized tumors and metastatic tumors by FDG PET/CT followed by the serum metabolites of the NMR studies and tissue Western blotting. The validations of localized and metastatic tumors were by histology.

**Figure 3 metabolites-13-00110-f003:**
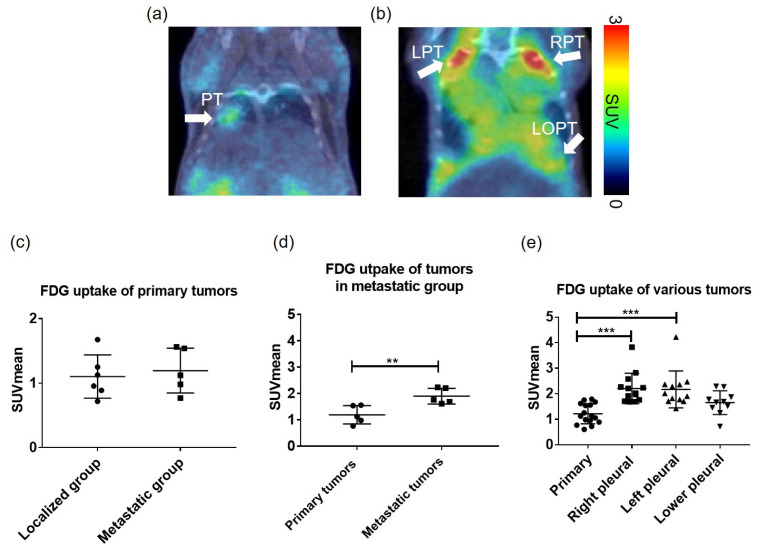
18F-FDG uptake in primary lung tumors and metastatic tumors. (**a**) Representative coronal 18F-FDG PET/CT fused image showed one primary lung tumor (PT) growing in the left lung parenchyma in the localized group. (**b**) Representative coronal 18F-FDG PET/CT fused images of the metastatic group. In the left lung region, the left pleural tumor (LPT) was considered an invasion of the primary tumor. In the right lung region, the right pleural tumor (RPT) and lower pleural tumor (LOPT) were defined as metastases. (**c**) The 18F-FDG uptakes of primary lung tumors between localized and metastatic mice show no significant difference (localized group, SUV_mean_: 1.10 ± 0.34, N = 6, vs. metastatic group, SUV_mean_: 1.20 ± 0.35, N = 5, *p* > 0.79). (**d**) Tumor 18F-FDG uptake in metastatic mice, the metastases significantly elevated compared with primary tumors (primary tumors, SUV_mean_: 1.20 ± 0.35 vs. metastatic tumors, SUV_mean_: 1.97 ± 0.30 **, N = 5 for each group, ** *p* < 0.01. (**e**) Collected various lung tumors in studied mice, the significantly increased 18F-FDG uptake of tumors in the pleural was found (primary tumors, SUV_mean_: 1.21 ± 0.39, N = 16 vs. right pleural tumors, SUV_mean_: 2.21 ± 0.60 ***, N = 13, left pleural tumors, SUV_mean_: 2.17 ± 0.72 ***, N = 12, respectively ** *p* < 0.01, *** *p* < 0.001).

**Figure 4 metabolites-13-00110-f004:**
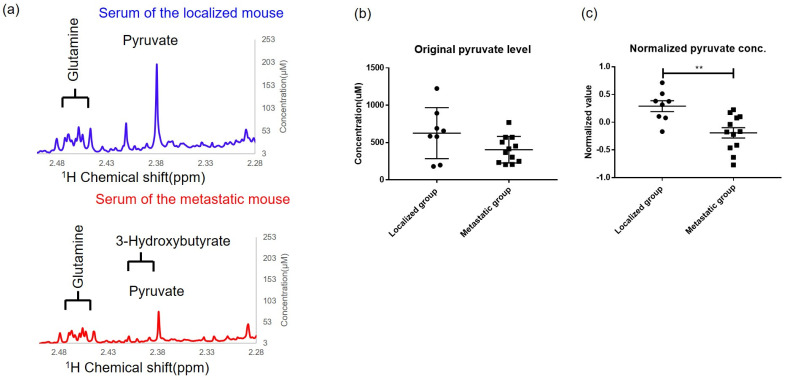
Serum metabolites of tumor-bearing mice analyzed by NMR and MetaboAnalyst. (**a**) The comparison of the representative certain serum metabolites of NMR spectrums in a localized mouse and metastatic mouse were displayed. The serum pyruvate level was decreased in the metastatic mouse. (**b**) There was a slightly lower original pyruvate level in the serum of the metastatic group (localized group, pyruvate level: 626.2 ± 121, N = 8 vs. metastatic group, pyruvate level: 404.2 ± 51.12, N = 12, *p* = 0.07). (**c**) The pyruvate serum levels were processed and normalized by median, log transformation, and Pareto scaling in data processing in MetaboAnalyst software. The significantly lower pyruvate level in serum of the metastatic mice (localized group, N = 8, normalized pyruvate level: 0.29 ± 0.27 vs. metastatic group, N = 12, normalized pyruvate level: −0.19 ± 0.32 **, respectively, ** *p* < 0.01).

**Figure 5 metabolites-13-00110-f005:**
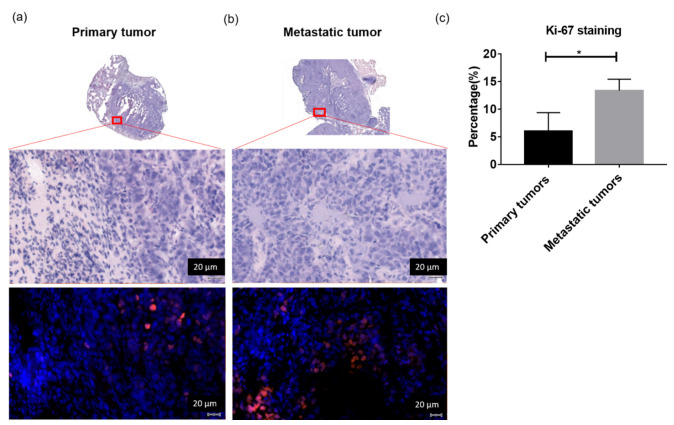
Metastatic tumors showed a higher proliferation rate compared with primary tumors that are from the localized group. (**a**) Representative HE-stained sections of primary tumors infiltrated by cells with large nuclei corresponding to Ki67-stained and DAPI-stained sections of primary tumors (×200, scale bars 20 µm). (**b**) Representative HE-stained sections of metastatic tumors with disturbed cells corresponding to Ki67-stained and DAPI-stained sections of metastatic tumors (×200, scale bars 20 µm). (**c**) The quantitative expression of fluorescent Ki-67 of metastatic tumors is significantly increased compared to that of primary lung tumors (primary tumors 6.052 ± 1.934% vs. metastatic tumors 13.41 ± 1.184%, N = 3 for each group, * *p* < 0.05).

**Figure 6 metabolites-13-00110-f006:**
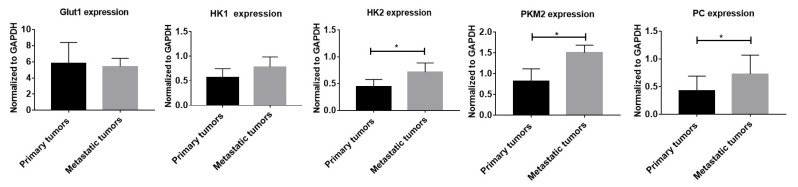
The glycolytic protein expression in primary tumors from the localized group and metastatic tumors from the metastatic group by Western blotting studies. Quantification of the expression of Glut1, HK1, HK2, PKM2, and PC protein by normalized to GAPDH expression. The significantly increased expression of HK2, PKM2, and PC protein in the metastatic tumors was shown. N = 3 in each group for Glut1, HK1 and HK 2, N = 4 in each group for PKM2, N = 5 in each group for PC, * *p* < 0.05.

**Figure 7 metabolites-13-00110-f007:**
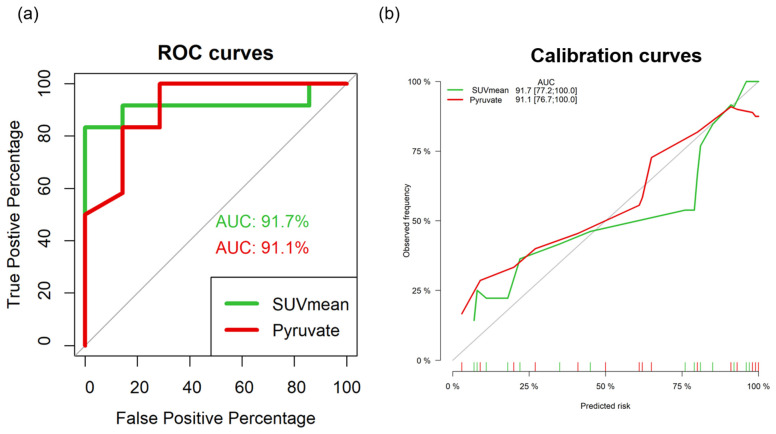
ROC curves and calibration curves of PET/CT parameters, SUV_mean_, and serum metabolites, pyruvate to diagnose the localized (N = 7) or metastatic (N = 12) diseases. (**a**) The imaging parameter, SUV_mean_ shows slightly higher AUC values than that of serum metabolite, pyruvate; however, the AUCs of the two parameters are over 90%, SUV_mean_ and pyruvate are the promising predictors for the differentiation of localized or metastatic lung cancers. (**b**) The calibration curve of imaging parameter and serum metabolite, normalized pyruvate level in a preclinical lung cancer model.

**Table 1 metabolites-13-00110-t001:** The significantly different expression of various serum metabolites from the localized mice and metastatic mice were listed.

Metabolites	VIP Score >1.5	Fold Change (Metastasis/Localized) >1.2 or <0.8)	*p*-Value
Pyruvate	2.3382	0.65992	0.004294 **
O-phosphocholine	1.8351	1.4604	0.057379
Phenylalanine	1.7429	0.75095	0.057379
Creatine	1.7341	0.79178	0.13494
Fumarate	1.5474	0.79517	0.34315

Metastatic group vs. localized group, statistically significant, ** *p* < 0.01.

## Data Availability

The datasets used and/or analyzed during the current study are available from the corresponding author on reasonable request. Data is not publicly available due to privacy.

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
