# Peer review of "Glycolytic Plasticity of Metastatic Lung Cancer Captured by Noninvasive 18F-FDG PET/CT and Serum 1H-NMR Analysis: An Orthotopic Murine Model Study"

_metabolites, 2023, doi:10.3390/metabo13010110_

Round 1

Reviewer 1 Report

The manuscript ID metabolites-2134246 mainly presents a study about a particular model related to noninvasive diagnostic system to capture early phenotypic transformation for metastasis using 18F-FDG PET and 1H-NMR-based serum metabolomics. Standard characterization was carried out and consistent results were obtained. However, some important issues are present in the manuscript that in my opinion should be addressed. Please see below a list of comments to the authors.

1. The selection of several parameters in section 2.5 should be justified and explained.

2. The incorporation of the calibration curves in the manuscript to guarantee the sensitivity reported in figure 6 would be useful for readers.

3. A photo of a representative experiment fo this study would be welcome.

4. For improving the accuracy and validation of statistical measurements, advanced techniques like machine learning for biosensing can improve the presentation of experimental results. The authors are invited to briefly discuss about the perspective of this tool in this topic and see for instance: https://doi.org/10.3390/bios12090710

5. The authors should clearly highlight what this report adds to literature in respect to updated comparative works. You can see for instance: DOI: 10.1016/j.cllc.2022.08.001 and https://doi.org/10.3389/fonc.2022.1005924

6. Advantages and disadvantages of the proposed platform should be described with better details.

7. It is suggested to split the collective citations in individual form in order to better justify the references selected for the presentation and discussion of the topic analyzed.

8. There is only one 2022 original articles in the list of references. The bibliography should be updated in some cases.

9. Please increase the fonts in the micrographs shown in figure 4.

10. Besides a proofreading is mandatory throughout the text, please complete the information of references. A clear example that should be supported is the information for reference 41.

Reviewer 2 Report

Chung et al. presented a study using FDG PET imaging and NMR analysis of serum metabolites to provide a better prognostic indicator for lung cancer in a murine model. The authors should address the following comments before further consideration for publication:

1.       In line 56, the authors said “FDG only partially provides macroinformation on glycolysis in primary tumors and metastasis”. The authors should elaborate on this point and provide relevant references.

2.       It’s not clear to me how the proton NMR of serum metabolite can complement the existing FDG imaging. The author themselves showed that FDG can already be used to distinguish primary and metastatic tumors in Figure 2. Maybe the authors can elaborate more on this point.

3.       In figure 3c, how are the pyruvate levels normalized?

4.       In figure 6, the authors showed good correlation between SUVmean and pyruvate level. This seems to be a rather obvious conclusion since both are the results of metabolism. Is there a case when the two don’t agree and we can use that to derive a different diagnosis?

5.       While it sounds reasonable to use serum metabolites NMR to facilitate prognosis, the authors failed to discuss the potential pitfalls of using this approach. For example, increasing glycolysis can be associated with various non-cancer related conditions such as inflammation. The authors should include discussion of potential pitfalls in their discussions.

Round 2

Reviewer 1 Report

The current reviewed version of the manuscript still presents some fundamental issues to be addressed, please see below:

*Regarding that the aim of the work deals with the proposing a noninvasive diagnostic platform, the selection of several parameters in section 2.5 should be clearly justified in order to see that the study is systematic instead of incidental.

*Moreover, the reference material employed for specifying the concentration in the calibration curve should be indicated. Specially if the calibration requires a comparative technique of measurement.

*The discussion of the work should be edited to be more attractive and specific for readers from the audience of this prestigious journal. This is in order that this work can be useful for future research and cited.

Round 3

Reviewer 1 Report

The manuscript can be considered for publication in present form.